# Effects of COVID-19 on trade flows: Measuring their impact through government policy responses

Javier Barbero[1☯], Juan José de Lucio[2☯]*, Ernesto Rodríguez-Crespo[3☯]

1 Joint Research Centre (JRC), European Commission, Seville, Andalucía, Spain, 2 Department of Economic Structure and Development Economics, Universidad de Alcalá de Henares, Alcalá de Henares, Madrid, Spain, 3 Department of Economic Structure and Development Economics, Universidad Autónoma de Madrid, Madrid, Spain

☯ These authors contributed equally to this work.
* Juan.deLucio@uah.es

**Data Availability Statement:** The data underlying the results presented in the study are available from UN Comtrade (https://comtrade.un.org), the Centre d'Études Prospectives et d'Informations Internationales (CEPII) Gravity database (http://

## Abstract

This paper examines the impact of COVID-19 on bilateral trade flows using a state-of-the-art gravity model of trade. Using the monthly trade data of 68 countries exporting across 222 destinations between January 2019 and October 2020, our results are threefold. First, we find a greater negative impact of COVID-19 on bilateral trade for those countries that were members of regional trade agreements before the pandemic. Second, we find that the impact of COVID-19 is negative and significant when we consider indicators related to governmental actions. Finally, this negative effect is more intense when exporter and importer country share identical income levels. In the latter case, the highest negative impact is found for exports between high-income countries.

## Introduction

The world is facing an unexpected recession due to the disruption of the COVID-19 pandemic in the global economy. In parallel with the consequences of the 2008–2009 crisis, international trade has once again collapsed. World trade volumes decreased by 21% between March and April 2020, while during the previous crisis the highest monthly drop was 18%, between September and October 2008. Cumulative export growth rate for the period December 2019–March 2020 was -7%, while for the period July 2008 –February 2009 it was -0,8%. The 2020 downturn was less prolonged than that caused by the latter crisis. Trade volumes in August 2020 only showed a 3% decrease compared to March 2020. The World Trade Organization (WTO) estimated that international merchandise trade volumes fell by 9.2% in 2020, a figure similar in magnitude to the global financial crisis of 2008–2009, although factors such as the economic context, the origins of the crisis and the transmission channels are deemed to be very distinct from the previous crisis [1].

Due to its rapid propagation, a proper evaluation of the economic impacts of COVID-19 crisis is not only desirable but challenging if the aim is to mitigate uncertainty [2]. The

www.cepii.fr/cepii/en/bdd_modele/presentation.
asp?id=8) and from Our World in Data COVID-19
Git Hub repository (https://github.com/owid/covid-
19-data/tree/master/public/data). The three
datasets are publicly available for all researchers.
Merging the three datasets and following the steps
described in the "Model description and estimation
strategy" section readers can replicate the results
of this manuscript.

**Funding:** de Lucio and Rodríguez-Crespo thank
financial support from Universidad de Alcalá de
Henares (UAH) and Banco Santander through
research project COVID-19 UAH 2019/00003/016/
001/007. De Lucio also thanks financial support
from Comunidad de Madrid and UAH (ref: EPU-
INV/2020/006 and H2019/HUM5761).

**Competing interests:** The authors have declared
that no competing interests exist.

COVID-19 crisis has its origins in the policy measures adopted to combat the health crisis, while the 2008–2009 crisis had economic roots contingent on financially related issues. At the current time, the collapse of international trade has been driven by the voluntary and mandatory confinement measures imposed on world trade. We aim to analyse the impact of said confinement measures on trade. Estimating COVID-19 impacts on trade would shed light on the cost of confinement measures and the evolution and forecast of bilateral trade.

From an empirical point of view, we resort to [3], who use quarterly data for the period 2003–2005 in order to analyse the impact of the SARS epidemic on firms. They show that regions with higher transmission of SARS experienced lower import and export growth compared to those in the unaffected regions. The propagation of a virus resembles natural disasters, with both interpreted as external non-economic based shocks, the effects of which have been addressed already in the literature (e.g., [4–9]).

Another strand of the literature directly analyses the effects of the COVID-19 pandemic on global trade in terms of the transmission mechanism of the shock: demand, supply and global supply chains. Some authors argue that demand factors have played an important role in explaining the shock [10, 11] conclude that both shocks (demand and supply) are present in the crisis [1] highlight the role of global value chains in the transmission of shocks [12] focus on supply chain disruptions and reveal that those sectors with large exposure to intermediate goods imports from China contracted more than other sectors [13]. Focus on the role of global supply chains on the GDP growth for 64 countries during the COVID-19 pandemic and show that one quarter of the total downturn is due to global supply chains transmission. They also conclude that in general global supply chains, make countries more resilient to pandemic-induced contractions in labour supply. Finally, the collapse in trade can also be considered as a trade-induced effect caused by economic recessions (e.g., [14–16]) and may also be associated with the impact of COVID-19.

During the current wave of globalization, time lags in synchronizing business cycles between countries are reduced significantly in terms of the intensity of trade relationships [17]. Find that business cycle synchronization increases when countries trade more with each other [18]. Show that bilateral trade intensity has a sizeable positive, statistically significant, and robust impact on synchronization. These results are in line with [19], who finds that greater trade intensity increases business cycle synchronization, especially in country pairs with a free trade agreement and among industrial country pairs. Our paper provides prima facie evidence that this relationship also holds during pandemic-related trade shocks.

We contribute to the literature by integrating monthly data for a trade analysis of 68 countries, 31 of which are classified as high-income. We additionally focus on differential effects between high-income and low- and middle-income countries. This paper aims to shed light on the impact of COVID-19 on exports by means of an integrated approach for a significant number of countries, thereby avoiding an individual analysis of a single country or region that could potentially be affected by idiosyncratic shocks. Given the existence of substantial differences in trade performance and containment measures exhibited by countries and trade partners and attributable in part to their income, we also study whether the impact of COVID-19 on trade differs in terms of income levels. To the best of our knowledge, both questions, the integrated impact of confinement measures and the income related effects, remain unexplored in previous studies.

A proper analysis of ex-post trade impacts related to COVID-19 requires a suitable and fruitful methodology. Gravity models can be helpful in achieving this goal, since they have recently started to incorporate monthly trade data into the analyses, albeit with empirical evidence that is still scarce and far from conclusive [8, 20, 21]. At the same time, several methodological issues need to be resolved adequately when using gravity models [22]. Resorting to

monthly data may pose several advantages in terms of accomplishing our research goals and exploiting the explanatory power exhibited by gravity models and monthly country confinement measures. First, the data reflect monthly variations and allow us to better capture any differential effects arising across countries. Second, annual trade data do not capture the short-run impact of shocks that occur very rapidly, something which a monthly time span can achieve. Monthly data can pick up any rapid movements associated with COVID-19 measures and allow for differential shocks in relation to months and countries. This is particularly relevant given the growing importance of nowcasting and short-term analysis techniques required nowadays for an understanding of world economy dynamics. Finally, monthly data can explain the relative importance of demand and supply shocks during the course of the trade crisis.

We collect monthly trade data for 68 countries (S1 Table), which exported to 222 destinations between January 2019 and October 2020. Using state-of-the-art estimation techniques for trade-related gravity models, our results are threefold. First, we reveal a negative impact of COVID-19 on trade that holds across specifications. Second, we obtain results that do not vary substantially when considering different governmental measures. Finally, our results show that the greatest negative COVID-19 impact occurs for exports within groups (high-income countries and low-middle-income countries), but not between groups. These findings are robust to different tests resulting from the introduction of lagging explanatory variables, alternative trade flows (exports vs imports as the dependent variable) or COVID-19 impact measures (independent variables such as stringency index or the number of reported deaths per million population).

## Literature review on COVID-19 and trade

The specific literature covering the COVID-19 induced effects on trade can be catalogued as flourishing and burgeoning, but also as incipient and inconclusive at the current time. Some studies have addressed the impact of the health-related crisis on trade. The first strand of literature analyses the effects of previous pandemics by emphasizing asymmetric impacts across sectors. Using the quarterly transaction-level trade data of all Chinese firms in 2003 [3], estimate the effects of the first SARS pandemic on trade in that year. They find that (i) Chinese regions with a local transmission of SARS experienced a lower decline in trade margins, and (ii) the trade of more skilled and capital-intensive products was less affected by the pandemic.

Despite data being scarce, other studies focus on the current COVID-19 trade shock but are usually restricted to specific countries. For the case of Switzerland [23], combine weekly and monthly trade data, for the lockdown between mid-March and the end of July. They use goods information disaggregated by product and trade partner. They find that: (i) During lockdown Swiss trade fell 11% compared to the same period of 2019, and this trade shock proved more profound than the previous trade shock in 2009, (ii) contraction in Swiss exports seems to be correlated with the number of COVID-19 cases in importing countries, but at the same time, Swiss imports are related to the stringency of government measures in the exporter country (iii) for the case of products, only pharmaceutical and chemical products remained resilient to the trade shock and (iv) the pandemic negatively affected the demand and supply sides of foreign trade [24]. Use a gravity model and focus on exports from China for the period January 2019 to December 2020. They find a negative effect of COVID-19 on trade, but said effect is largely attenuated for medical goods and products that entail working from home.

For the case of Spain [10], find that for the period between January and July 2020, stringency in containment measures at the destination countries decreased Spanish exports, while imports did not succumb to such a sharp decline. Finally [25], extends the discussion of the Spanish case to analyse the impact of COVID-19 on trade in goods and services, corroborating

the existence of a negative effect. He finds a more pronounced decline for trade in services, due to the importance of tourism in the Spanish economy.

Other studies have provided additional evidence by considering a larger sample of countries. Using monthly bilateral trade data of EU member states covering the period from June 2015 to May 2020 [20], use a gravity model framework to highlight the role of chain forward linkages for the transmission of Covid-19 demand shocks. They explain that when the pandemic spread and more prominent measures were taken, not only did demand decrease further, but labour supply shortage and production halted [21]. Find a negative impact of COVID-19 on trade growth for a sample of 28 countries and their most relevant trade partners. Their findings suggest that COVID-19 has affected sectoral trade growth negatively by decreasing countries' participations in Global Value Chains from the beginning of the pandemic to June 2020 [26]. Analyse the impact of COVID-19 on trade for a larger sample of countries, focusing on export flows for 35 reporting countries and 250 partner countries between January and August in both 2019 and 2020. However, they restrict, their study exclusively to trade in medical goods and find that an increase in COVID-19 stringency leads to lower exports of medical products. Finally [27], use maritime trade shipping data from January to June 2020 for different countries, such as Australia, China, Germany, Malaysia, New Zealand, South Africa, United States, United Kingdom, and Vietnam. By applying the automatic identification system methodology to observational data, they obtain pronounced declines in trade, albeit the effect is different for each country.

Surprisingly, little attention has been paid to the impacts of COVID-19 on trade for different country income levels and we find several reasons to consider this issue as important. First, the role of trade costs is important, as the latter are related to economic policy and direct policy instruments (e.g., tariffs) are less relevant compared to other components [28]. According to [29], differences are expected in trade between high-,low- and middle-income countries due to the composition of trade costs: information, transport, and transaction costs seem to be more important for trade between high-income countries, while trade policy and regulatory differences better explain trade between low- and middle-income countries. The second reason is related to the composition of products, given that average skills in making a product are more intensive in high-income countries resulting in increasing complexity of the products traded compared to low- and middle-income countries [30]. Accordingly, specific product categories incorporate more embedded knowledge and their production may require engaging in a global production network with multiple countries. However, it has been alleged that participation of countries in global value chains depends on their income levels due to the objectives pursued: high-income countries focus on achieving growth and sustainability, while low- and middle-income countries seek to attract foreign direct investment and increase their economic upgrading [31]. Third, it is found that low-income countries present a lower share of jobs that can be done at home [32], rendering them more sensitive to lockdowns that affect services. Finally, due to the paucity of health supplies to mitigate COVID-19, as certain healthcare commodities may not be affordable for certain low-income countries [33]. Consequently, the effect of COVID-19 on the global economy may be more pronounced for those countries with fewer healthcare resources and impacts on trade do not constitute an exception.

Due to the reasons mentioned above, we expect different countries' responses to trade shocks induced by COVID-19 depending on their income levels, but this issue remains largely unexplored by the academic literature. The only exception is [34], but they reduce their analysis to COVID-19 impacts on trade concerning Commonwealth countries. Using the period from January 2019 to November 2020, they find ambivalent evidence: an increase in the number of COVID-19 cases in low-income countries reduced Commonwealth exports, but an identical scenario in high-income countries boosted their export flows.

All these findings are summarized as follows. In Table 1, we present a compilation of studies using monthly data that feature the impact of COVID-19 on trade.

It is also worth noting that the main shock started in March 2020, when most countries closed their borders and implemented lockdown measures. Accordingly, further analysis of COVID-19 impacts on trade requires periods with high frequency, such as monthly data, in order to deliver satisfactory results. Apart from the aforementioned studies in the context of COVID-19 research, to the best of our knowledge, monthly data are scarcely used in gravity models. They have been used either in the context of trade preferences [35, 36] or, more recently, to study the impact of natural disasters on trade [8].

Additionally, we aim to provide evidence concerning the COVID-19 effects on trade at country level, without restricting our research to either specific countries or territories, or, specific trade flows, such as intermediate goods. Since the COVID-19 crisis is still ongoing, it is also necessary to incorporate the most recent and updated time spans to provide policy recommendations aligned with the business cycle. We intend to analyse whether COVID-19 impacts on trade have affected the world economy from a global perspective. This analysis will allow us to distinguish different impacts in terms of levels of economic development, which to the best of our knowledge, remain largely unexplored by the academic literature.

## Empirical analysis

This section is organized into three separate sub-sections. First, we describe the empirical model and the estimation strategy. Second, we report information on data issues. Finally, we cover country policy responses to COVID-19.

### Model description and estimation strategy

For the purpose of accomplishing our research objectives, we resort to a bilateral trade gravity model, which has progressively become the reference methodology for analysing the causal impacts of specific variables on trade (e.g., [22, 37–40]; among other scholars).

In this case, we aim to estimate the impact of COVID-19 on bilateral trade flows. For this reason, we depart from the functional form specified by [41], which can be augmented conveniently with variables related to COVID-19, as shown by [42] for the case of tourism flows. Our baseline gravity Eq (1) is augmented with COVID-19, and a set of geographical, economic, and institutional determinants that are commonly used in gravity models. This equation is shown in exponential functional form as follows:

$$X_{ijm} = \exp(\beta_0 + \beta_1 COVID_m + \beta_2 \ln DIST_{ij} + \beta_3 CONTIG_{ij} + \beta_4 COMLANG_{ij} + \beta_5 COLONY_{ij}$$
$$+ \beta_6 RTA_{ij} + \varphi_{im} + \gamma_{jm}) \times \varepsilon_{ijm} \tag{1}$$

**Table 1. A compilation of studies covering COVID-19 impacts on trade using monthly data.**

|  | Countries | Ending | Gravity |
|---|---|---|---|
| **Büchel et al. (2020)** [23] | Switzerland | July | No |
| **Kejzar and Velic (2020)** [20] | 27 EU | May | Yes |
| **Espitia et al. (2021)** [21] | 28 EU | June | Yes |
| **Hayakawa and Kohei (2021)** [26] | 35 world | August | Yes Medical products |
| **Khorana et al. (2021)** [34] | Commonwealth | November | Yes |
| **Minondo (2021)** [25] | Spain | - | No |
| **Ornelas et al. (2021)** [24] | China | December | Yes |
| **Verschuur et al. (2021)** [27] | 9 countries | June | No |
| **This study** | 68- Different income levels | October | Yes |

Where subscripts $i$, $j$ and $m$ refer to exporter and importer country and month, respectively. $COVID_m$ is a control variable that takes value 1 for a COVID-19 trade shock, after March 2020, and 0 otherwise. $DIST_{ij}$ is the geographical distance between exporter and importer country. $CONTIG_{ij}$ is a control variable that takes value 1 when exporter and importer country are adjacent and 0 otherwise. $COMLANG_{ij}$ is a control variable that takes value 1 when exporter and importer country share a common language and 0 otherwise. $COLONY_{ij}$ is a control variable that takes value 1 when exporter and importer country share past colonial linkages and 0 otherwise. $RTA_{ij}$ is a control variable that takes value 1 when exporter and importer country have a regional trade agreement in force and 0 otherwise. In addition to these explanatory variables, we also consider other control variables to capture omitted factors. $\varphi_{im}$ and $\gamma_{jm}$ are exporter-month and importer-month fixed effects, respectively. Finally, $\varepsilon_{ijm}$ stands for the error term.

The logic behind including these variables is found in the literature, and the explanation is provided as follows: COVID-19-related variables are introduced to estimate the impact of the current COVID-19-shock on trade, since it is expected to be detrimental (e.g., [3, 23]). Governmental actions are expected to reduce the duration and magnitude of COVID-19 shock by facilitating a smoother transition to a post-pandemic scenario while generating an economic downturn in the short-run due to the limitations of economic activity and the increase in government expenditure. Adjacency and distance relate to the geographical impacts on trade, given the great influence exerted by geography on trade patterns [43]. Adjacency is included due to the existence of a border effect, where countries tend to concentrate their trade flows with nearby trade partners [41, 44], so that higher adjacency leads to increasing trade flows. The reasons to include distance in gravity models stem from the early contributions [45]. Countries prefer to trade with less distant trade partners, so that a negative coefficient is expected. Colonial linkages and common language respond to the flourishing literature on the impact of institutions on trade, where the latter play a key role in reducing trade costs and facilitating trade (e.g., [46, 47]; among others). Finally, regional trade agreements have multiplied exponentially in the context of globalization and trade liberalization, so that they contribute to decreasing trade costs and enhancing trade [48, 49].

Exporter-month and importer-month fixed effects are included to comply with multilateral resistance terms (MRTs), which are related to third-country impacts on the bilateral relationship. They are considered as a pivotal element of modern gravity equations [41]. According to the structural gravity literature, the omission of such aforementioned MRTs is expected to lead to inconsistent and biased outcomes [22].

Concerning the previous gravity specification, it is important to highlight that our baseline gravity Eq (1) does not contain GDP, which is considered a fundamental variable in the seminal gravity models because it measures country size [45]. The omission of GDP is intentional due to several reasons: (i) GDP variables tend to vary quarterly or yearly and (ii) the inclusion of exporter-month and importer-month fixed effects are perfectly collinear with GDP, so that these control variables will capture its effects on trade.

We estimate the gravity model using the pseudo-poisson maximum likelihood (PPML) estimator. The PPML estimator has been widely used in the gravity literature, given that it removes the heteroskedasticity from taking logarithms of trade flows and allows the inclusion of zeros in the regression, thus providing a consistent estimation [50]. Another advantage of using PPML with exporter-time and importer-time fixed effects is that the MRTs can be captured properly in order to comply with structural gravity theory [22]. However, the inclusion of COVID-19 government response indices, which are exporter-month and importer-month varying, requires the modification of our baseline Eq (1) to avoid perfect collinearity with the set of fixed effects. To accomplish the foregoing we consider an alternative Eq (2), which is

shown as follows in multiplicative form:

$$X_{ijm} = \exp[\beta_0 + \beta_1(COVID_{im} * RTA_{ij}) + \varphi_{im} + \gamma_{jm} + \eta_{ij}] \times \varepsilon_{ijm} \tag{2}$$

Eq (2) introduces some novelties in relation to the previous Eq (1). First, by interacting the COVID-19 variable with the control variable for regional trade agreements we can compute the impact of COVID-19 on trade by assuming that countries with regional trade agreements affect this empirical relationship. Thus we assess whether COVID-19 impacts on trade are more (less) profound for these countries with (without) previous regional trade agreements. We sum 1 to the variable before taking the logs to avoid losing the observations before the COVID started. This strategy responds to the strands of literature that acknowledge the role of international trade as a driver of business cycle synchronization (e.g., [51, 52]) and, more specifically, that regional trade agreements may be behind the transmission of shocks across countries (e.g., [19, 53, 54]). In particular, we follow [26] approach since the interaction between COVID-19 variables and regional trade agreements allows us to study the heterogeneous impacts of COVID-19 on trade by bringing economic linkages into the discussion.

Another advantage is that interacting an explanatory variable with a control variable may also relieve us from endogeneity issues, as shown by [55]. In particular, COVID-19 impacts on trade may be driven by omitted variable bias. However, this comes at the cost of not interpreting exporter and importer impacts simultaneously, as both coefficients become symmetric and only one of them can be interpreted, as shown by [56] and [57] when studying the impact of institutions on trade using gravity equations at country and regional level, respectively.

Eq (2) also considers a third set of fixed effects, which are exporter-importer pair effects denoted by $\eta_{ij}$. Considering pair effects in the gravity equation as an explanatory variable may pose the advantage of mitigating the estimation from endogenous impacts induced by time-invariant determinants (e.g., [58, 59]) and hence may improve the empirical specification. Three-way fixed effects, constituted by pair, exporter-month and importer-month fixed effects, have become the spotlight in gravity specifications assessing the impact of natural disasters on trade, even for those scholars using monthly data, such as [8].

Finally, we acknowledge that the simultaneous inclusion of such three-way fixed effects, requires large amounts of data in order to carry out the estimation procedure. For this reason, sophisticated PPML estimation commands have recently been developed for gravity equation estimations, so that they can include a high number of dimensional fixed effects and run relatively fast in contrast to previous existing commands [60, 61].

## Data

Our sample covers a set of 68 countries exporting across 222 destinations, between January 2019 and October 2020 with 31 of these exporters classified as high-income countries. Due to the specific monthly nature of COVID-19 shock, we rely on monthly bilateral trade flows gathered from UN Comtrade. Trade data were extracted on the 15 of February 2021, using the UN Comtrade Bulk Download service. According to the degree of availability of monthly trade flows for countries, our analysis covers aggregate trade flows. For those observations with missing trade flows, we conveniently follow previous studies that suggest that missing trade flows can be completed with zeros, [50, 62].

Variables related to the COVID-19 government response have been taken from the systematic dataset of policy measures elaborated by the Blavatnik School of Government at Oxford University [63]. These indices refer to government response, health measures, stringency, and economic measures. Their composition and implications are described more broadly in the following sections.

The rest of the variables, institutional and geographical, are gathered from the Centre d'Études Prospectives et d'Informations Internationales (CEPII) Gravity database [64]. S1 and S2 Tables show the list of countries and the main descriptive statistics, respectively.

### Policy responses to COVID-19: equal or unequal?

Once COVID-19 spread across a significant number of countries, they were urged to implement policy actions of response. For this reason, several indicators were built to measuring countries´ governmental response to COVID-19. As mentioned previously, the set of policy indicators developed by [63] constitutes the most noteworthy approach for measuring countries´ policy responses. The four indicators are described as follows:

- Stringency index (upper left chart, Fig 1) contains the degree of lockdown policies to control the pandemic via restricting people´s social outcomes. The index is built using data on the

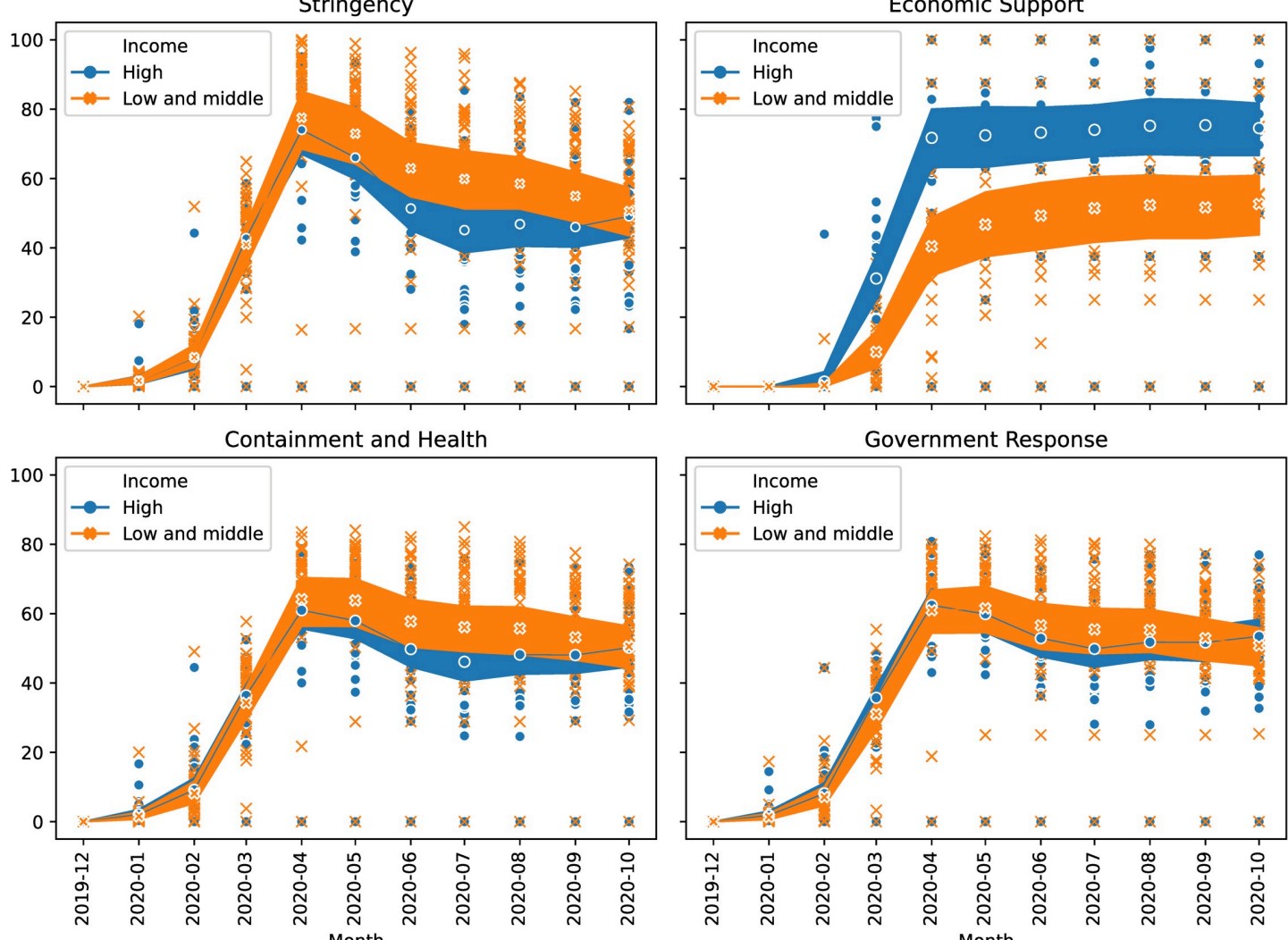

**Fig 1. Evolution of the four COVID-19 policy indicators between December 2019 and October 2020 by income level for our sample of exporting countries.** Source: own elaboration from [63]. Note: each point represents a country, and the concentration of countries with similar values produces darker areas. Additionally, the mean and 95% confidence bands are represented.

closure in education (schools and universities), public transport and workspaces, the cancel-
lation of public events, limits on gatherings, restrictions in internal movements, and orders
to confine at home.

- Economic Support index (upper right chart, Fig 1) includes measures related to public
expenditure, such as income support to people who lose their jobs or cannot work, debt relief
to households, fiscal measures, and spending to other countries.

- Containment and Health index (lower left chart, Fig 1) combines lockdown measures with
health policies, such as the openness of the testing policy to all the population with symptoms
or asymptomatic, the extent of the contact tracing, the policy on mandatory use of facial cov-
erings, and monetary investments in healthcare and in vaccines.

- Overall Government Response index (lower right chart, Fig 1) collects all governments'
responses to COVID-19 by assessing whether they have become stronger or weaker. This
index combines all the variables of the containment and health index and the economic sup-
port index.

These indices vary from 0 to 100, with a higher value indicating stronger country measures
in response to COVID-19. As these indicators are collected daily, we convert them to monthly
averages. The evolution of the four indicators is presented in Fig 1, where each point repre-
sents a country and the concentration of countries with similar values produces darker areas.
Additionally, the mean and 95% confidence bands are represented. We pay special attention to
differences between high-income and low- and middle-income countries, in line with our
research objectives.

Stringency reaches its maximum in April 2020 when the first wave reaches its peak in most
countries. Since then, restrictions have been slowly lifted but started to increase again after the
summer in high-income countries, coinciding with the beginning of the second wave.

Economic support increased rapidly in March and April and remains stable, with high-
income countries granting more economic support to the population than low- and middle-
income countries. The economic support variable identifies significant differences between
high-income and low- and middle-income countries during the whole period.

The containment and health index and the overall government response index, present a
similar pattern regarding income levels. However, we observe that low- and middle-income
countries relaxed the measures from April 2021 onwards, whereas high-income countries did
so in July 2021. In any case, the countries analysed show significant variability in all the indi-
ces, as indicated by the estimates made in the following section.

To sum up, we find that in low- and middle-income countries, pandemic measures have
been slightly stricter than in high-income ones, as the values of their COVID-19 policy
responses indices are higher for all cases except for the economic support index. The greater
availability of resources in high-income countries to control the pandemic explains this
difference.

Fig 2 displays the evaluation of total monthly exports in 2020, relative to January 2020, by
income level for our sample of exporting countries. We observe the big decline in exports
between March and April mentioned previously. In fact, the observed magnitude of trade
decline as a consequence of COVID-19 is identical to the previous global recession, but con-
tractions in GDP and trade flows are more profound at the current stage [65]. However, we
observe that high-income countries have gradually been recovering their export flows, reveal-
ing a larger degree of resilience and how economic support policies might have helped them in
recovering economic activity. In particular, greater firm engagement in trade because of

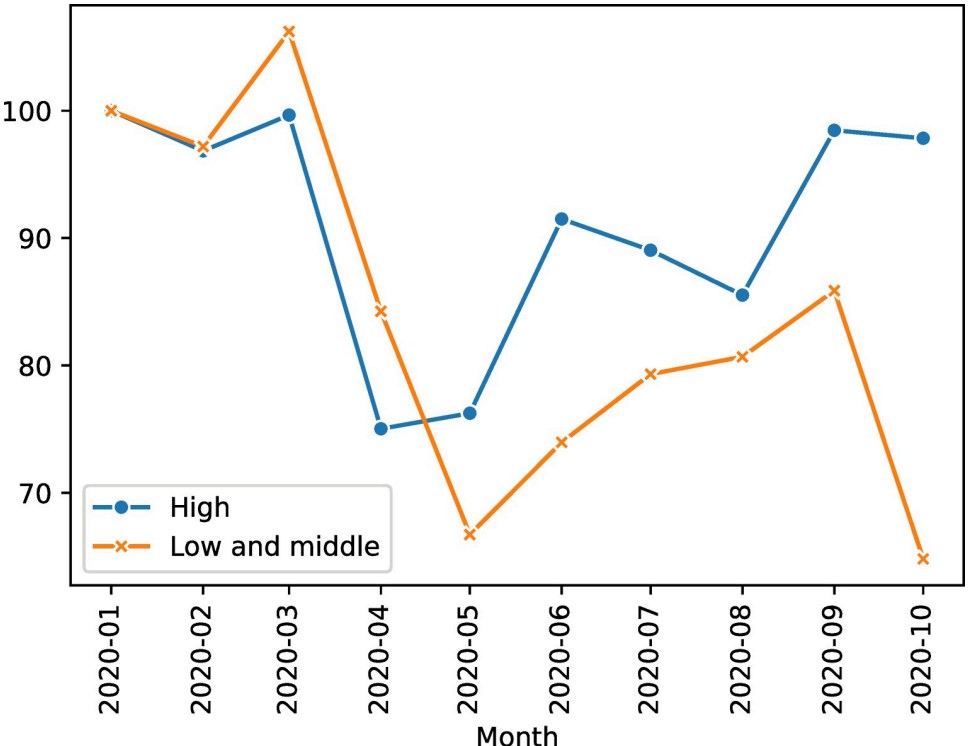

**Fig 2. Monthly evolution index of total exports in 2020, relative to January 2020 = 100.** Source: own elaboration using UN Comtrade trade data.

previous global recession may be beneficial, as they have been able to recover in a shorter period of time from this new contraction in GDP due to the openness to foreign markets [66]. Find that net firm entry in export markets contributed less to export growth during the Great Trade Collapse, between 2008 and 2013, than continuing exporters. Export capacity to foreign markets in order to counteract the negative impact of local demand shocks is illustrated by [67] for the specific case of Spain.

We acknowledge that policy responses differ by country, as the impacts of COVID-19 have been strongly unequal for countries due to several reasons. First, countries have reported differences in the number of deaths, mainly attributable to the population composition. There is an increasing number of elder populations in a significant number of high-income OECD countries [68] and this group is the most vulnerable to COVID-19 (e.g., [69]).

At the same time, countries have also implemented trade policy actions to mitigate the influence of COVID-19 on the global economy. For the sake of brevity and in line with the aim of the article, we only consider the trade policy response. Readers interested in analyzing a complete set of economic policy responses (i.e., budgetary and/or monetary) to COVID-19 are entitled to check the International Monetary Fund (IMF) Policy Tracker at https://www.imf.org/en/Topics/imf-and-covid19/Policy-Responses-to-COVID-19.

We focus on the corresponding period coincident with our sample, but further trade policy actions have been implemented after this period because of the increasing number of cases related to COVID-19 in subsequent waves. They can be checked at the IMF Policy Tracker previously reported, which is updated regularly [70]. Summarizes the major stylized facts during the first nine months of the pandemic. First, there was a noticeable rise in trade policy activism consisting mainly of export controls and import liberalization measures with strong

cross-country variation. Second, this activism was reported to vary by country and products, where medical and food products experimented a substantial overall increase in their demand from February 2020. Third, we observe a further trade liberalization process after May 2020, where the number of liberalization measures exceeded the number of trade restrictions in medical products.

Such cross-country variation in trade policy response aligns with our expectations since, as mentioned previously, trade specialization differs by country. Accordingly, their sensitivity to the growing demand for food and medical products may vary substantially. For this reason, some countries were more resilient to COVID-19 trade shocks than other countries, as shown by the decreases observed in their trade flows. To this end, we compare trade drops for the most affected countries relative to January 2020 and their governmental response, from May 2020 to October 2020. As described by [70], countries experienced a substantial relaxation in most of their trade measures in May 2020.

For the ten countries with the largest trade drop evidence is ambivalent. For the sake of brevity, we have omitted the data concerning each country and provide a general overview. Data is available from the authors upon request to the corresponding author and can be obtained from the UN Comtrade database. On the one hand, four high-income (Macao, Mauritius, Portugal, and Slovakia) and six middle-income countries (El Salvador, Mexico, Montenegro, Guyana, Egypt, and Romania) were among the most affected countries in May 2020, with El Salvador registering the highest level of governmental response. Trade relative to January 2020 ranges from 51 to 69 percent in this period. On the other hand, we find that the number of high-income countries increased to six in October 2020 but, at the same time, differences in governmental response decreased their observed variance. Israel registered the highest level of governmental response during this month. In this case, relative trade ranges from 76 to 102, corroborating the previous finding that countries recovered rapidly from this trade shock. To sum up, despite differences in governmental response due to the impact of COVID-19 by countries, recovery can be alleged to follow similar patterns for the most affected countries.

## Results

This section is organized into three separate sub-sections. First, we present a benchmark analysis, and afterwards we show the main results obtained for the four different COVID-19 government policy response indices. Finally, we review whether COVID-19 impacts differ by levels of economic development.

### Benchmark analysis

This section reviews four specifications of the gravity equation: Column (I) in Table 2 includes the COVID-19 binary variable without interactions and only including exporter and importer fixed effects. Column (II) departs from Eq (2) but introduces exporter-month and importer-month fixed effects and, finally, column (III) adds pair fixed effects to the specification showed in column (II). For this robustness analysis, we use $COVID_m$ variable, which takes value 1 from March 2020, when several countries worldwide implemented lockdown measures, and 0 otherwise.

The coefficients shown in column (I) may be biased because the specification does not accurately capture those factors related to MRTs and structural gravity. We hence move on to the alternative specification considered in columns (II) and (III), where we choose column (3) because it corresponds to Eq (2) and solves limitations in the baseline Eq (1).

**Table 2. Robustness of gravity estimators estimated by PPML, January 2019-October 2020.**

| Column | (I) | (II) | (III) |
|---|---|---|---|
| Dependent variable | Exports | Exports | Exports |
| COVID-19 shock | -0.204*** | | |
| | (0.016) | | |
| COVID-19 shock * RTA | | -0.061 | -0.050*** |
| | | (0.041) | (0.015) |
| Log of Distance | -0.729*** | -0.728*** | |
| | (0.010) | (0.009) | |
| Contiguity | 0.326*** | 0.325*** | |
| | (0.027) | (0.025) | |
| Common Language | 0.322*** | 0.322*** | |
| | (0.029) | (0.028) | |
| Colonial Linkage | 0.684*** | 0.690*** | |
| | (0.037) | (0.033) | |
| RTA | 0.711*** | 0.729*** | |
| | (0.021) | (0.021) | |
| Constant | 25.304*** | 25.279*** | 20.675*** |
| | (0.078) | (0.070) | (0.004) |
| Exporter controls | Yes | No | No |
| Importer controls | Yes | No | No |
| Time controls | No | No | No |
| Exporter-time controls | No | Yes | Yes |
| Importer-time controls | No | Yes | Yes |
| Pair controls | No | No | Yes |
| Observations | 171,116 | 160,740 | 168,348 |
| Pseudo R2 | 0.911 | 0.924 | 0.993 |

Notes: Robust standard errors in parentheses, such as

*** p<0.01

** p<0.05

* p<0.10.

Alternative specifications show an unequivocal detrimental effect of COVID-19 shock on trade. While the magnitude of the COVID-19 coefficient in column (I) is larger, -0.204, the size decreases when we move to our baseline specification in column (III), where we interact with the RTA variable, with an estimated coefficient of -0.050 when we include pair fixed-effects. The rest of the variables show the expected coefficients according to the theoretical insights and projections of gravity models.

## Results by COVID-19 government policy responses indices

We now estimate our results introducing the four COVID-19 government response indices using Eq (2), as governments in countries more affected by the pandemic are expected to response with stringency, health, and economic support measures. Table 3 shows that the effect of COVID-19 on trade is negative and significant for all the variables considered. We agree with the existing literature on the negative impact of COVID-19 on trade (e.g. [21, 23]); and also with the negative impact of previous pandemics [3]. We also find that results do not vary substantially across indices related to COVID-19, as they range between -0.009, for containment and health measures, and -0.012 for economic support. Although estimated

**Table 3. Results by COVID-19 government response indicator estimated by PPML, January 2019-October 2020.**

| Column | (I) | (II) | (III) | (IV) | (V) |
|---|---|---|---|---|---|
| **Dependent variable** | **Exports** | **Exports** | **Exports** | **Exports** | **Exports** |
| **COVID-19 shock** | -0.050*** | | | | |
| | (0.015) | | | | |
| **Stringency * RTA** | | -0.009** | | | |
| | | (0.004) | | | |
| **Economic Support * RTA** | | | -0.012*** | | |
| | | | (0.004) | | |
| **Containment & Health * RTA** | | | | -0.009** | |
| | | | | (0.004) | |
| **Government Response * RTA** | | | | | -0.010*** |
| | | | | | (0.004) |
| **Constant** | 20.675*** | 20.672*** | 20.674*** | 20.673*** | 20.673*** |
| | (0.004) | (0.004) | (0.004) | (0.004) | (0.004) |
| **Observations** | 168,348 | 168,348 | 168,348 | 168,348 | 168,348 |
| **Pseudo R2** | 0.993 | 0.993 | 0.993 | 0.993 | 0.993 |

Notes: Robust standard errors in parentheses, such as

*** p<0.01

** p<0.05

* p<0.1.

All the specifications include exporter-month, importer-month and pair fixed effects.

parameters are not statistically different from each other, this might indicate that countries demanding more support to boost their economies have been the most affected ones by the COVID-19 trade shock. We also test our results using the traditional variable of COVID-19 reported deaths per million population as impact measure of the pandemic by country. Results, available under request, show no relevant variation to those presented in the article, which might be considered as an additional robustness test of the results presented.

Our results suggest that COVID-19 may be detrimental to trade flows for those countries engaged in previous regional trade agreements compared to the countries that were not members of these agreements, as shown by the result of interacting these variables. However, interaction terms fail to reflect that those countries not participating in regional trade agreements were not affected by COVID-19, given the large set of existing possibilities of trade integration between countries. Furthermore, these countries could have been expanding their trade flows via preferential trade agreements, which are less restrictive than regional trade agreements.

Although the estimated elasticity may mistakenly appear low, it reflects a large elasticity of trade to COVID-19, given the observed change in the four indicators under consideration. For instance, the overall government response indicator increases, on average, from 3.16 to 70.09 from February to April 2020. This change corresponded to a 2,155% increase in the government response indicator that, multiplied by the estimated elasticity of -0.010, results in a sharp decrease in export flows of around 21%. The explanation may be twofold. On the one hand, the COVID-19 trade shock may be expected to be less dampening for the economy in comparison to the trade shock induced by the global financial crisis [23], as we highlighted previously. On the other hand, the COVID-19 trade shock is still ongoing and it may be necessary to include results for the second wave commencing September 2020.

We include two additional robustness tests. First, in S3 Table we show our results with lagged independent variables, in order to check whether there exist non-contemporary impacts

of COVID-19 on trade. Our results show that the estimated coefficients remain significant and with greater values than those presented in Table 3. Second, in S4 Table we consider the estimations for import trade flows as the dependent variable. The reason for including imports responds to [71] suggestion of using mirrored datasets, given that import trade flows are more subject to trade barriers than exports. In contrast to import trade regimes, most export trade regimes tend to be free and do not require additional documents or licenses to trade the goods. Results remain invariant in relation to those presented in Table 3 and S3 Table.

## Results by levels of economic development

Finally, we complement the results by analysing whether COVID-19 impacts on trade depend on the levels of economic development of exporter and importer countries, distinguishing between high, low and middle-income importers, we follow the last version of the World Bank ´s classification. These results are shown in Table 4, where each cell contains the estimated coefficient and robust standard error for a different estimation of the COVID-19 government response indicators in a PPML regression that includes exporter-month, importer-month, and pair fixed effects. For instance, column (I) presents results for trade between high income countries, where the first row shows the estimated parameter for our reference equation, in line with column (4) in Table 1. The following four rows correspond to the estimated parameters in Table 2 for COVID-19 related variables. Results with lagged independent variables, presented in S5 Table, show that estimated coefficients remain statistically significant and higher than those presented in Table 3.

We find that the COVID-19 effect on trade remains negative, but it seems to be inversely related to the income levels of the importer country. While the COVID-2019 dummy variable

**Table 4. Results by income levels and COVID-19 indicators estimated by PPML, January 2019-October 2020.**

|  | Within groups | | Between groups | |
|---|---|---|---|---|
| Column | (I) | (II) | (III) | (IV) |
| Exporter | High | Low | High | Low |
| Importer | High | Low | Low | High |
| Dependent variable | $X_{ijm}$ | $X_{ijm}$ | $X_{ijm}$ | $X_{ijm}$ |
| COVID-19 shock | -0.103*** | -0.055*** | 0.014 | 0.059** |
|  | (0.036) | (0.016) | (0.017) | (0.026) |
| Stringency * RTA | -0.021** | -0.012*** | 0.006 | 0.012* |
|  | (0.009) | (0.004) | (0.004) | (0.007) |
| Economic Support * RTA | -0.025*** | -0.011*** | 0.002 | -0.003 |
|  | (0.008) | (0.004) | (0.004) | (0.008) |
| Containment and Health * RTA | -0.021** | -0.012*** | 0.005 | 0.012* |
|  | (0.009) | (0.004) | (0.004) | (0.007) |
| Government Response * RTA | -0.021** | -0.012*** | 0.005 | 0.012* |
|  | (0.009) | (0.004) | (0.004) | (0.007) |
| Observations | 22,727 | 48,397 | 79,499 | 20,508 |

Notes: The dependent variable is exports Robust standard errors are in parentheses, such as

*** $p<0.01$

** $p<0.05$

* $p<0.1$.

Each COVID-19 indicator is estimated with a different regression but paired in the same column for the sake of brevity. All the specifications include exporter-month, importer-month and pair fixed effects.

registers the highest negative impact, -0.103 for exports between high-income countries, (see column (I)), it is -0.055 between low-and middle-income countries, (see column (II)). The greater effect of the pandemic among integrated countries is in line with the results obtained by [20]. The effect found by [34] is the opposite, as COVID-19 in high-income countries spurs Commonwealth countries´ trade. However, their analysis focuses on a selected group of countries and distinguishes product categories so that the greater demand for medical products during the pandemic can explain these results. This fact reinforces the importance of including many countries in the sample since we can expect substantial differences in cross-country variation. We also find no significant effect of COVID-19 on the trade from exports of high-income to low-income countries in column (III) and a positive effect of exports from low-income to high-income in column (IV). These results remain similar when using lagged independent variables, presented in S5 Table. As robustness tests we include in S6 Table estimations for import trade flows. Results also remain invariant in relation to those presented in S3 Table.

These differences, within similar income groups and between groups, could be explained by the following. It is expected that countries more integrated into global supply chains and with greater business cycle synchronization are the most affected by trade collapse. We find that the existence of cross-country differences in capabilities is relevant since they determine countries' comparative advantage [72]. In this context, high-income countries tend to be associated with exports of high-quality goods since they charge higher prices [73] and require larger amounts of skills to produce these goods [74]. Hence, the set of products exported between high-income countries may be related to goods with a higher degree of economic complexity, which exhibit a greater degree of resilience to economic shocks. This difference may also be explained by the higher demand for medical products by high-income countries, since the first COVID-19 wave starting in February 2020 reached this group of countries first.

We find a marginally significant positive effect for exports from low- and middle-income countries to high-income countries. This shows that countries belonging to this income group might have found an opportunity to supply the high-income countries' markets, with whom they have regional trade agreements, these proving most affected by COVID-19 during the months in our sample. Low and middle-income countries are indeed progressively increasing their gains from trade because of greater exposure to globalization [75]; and this growth of trade flows from low and middle- income countries to high-income countries corroborates this evidence. Finally, not all kinds of products have been affected in the same magnitude. Evidence for outdoor goods can be found in [76]. Domestic consumption products, such as food, had a better performance during the pandemic, and low-income countries specializing in exporting these sorts of goods to high-income countries might have increased their exports to high -income countries.

## Conclusions

In this study, we shed light on how the current COVID-19 crisis affects trade flows for the world economy during the first wave of the pandemic. We apply a PPML estimator with three sets of fixed effects in consistency with the recent literature on gravity models. Using monthly trade data for a sample of 68 countries, we find a negative impact of COVID-19 on trade flows that it is greater for countries with RTA. In addition, we also find a negative impact for a set of four indicators related to government responses against COVID-19, although a substantial variation in the impact on trade of the different measures is not observable. Furthermore, our results show that the COVID impacts on trade are only negative when income levels for exporter and importer country with regional trade agreements are identical, and in particular for high-income level countries.

These results pose important policy recommendations. The current trade shock induced by COVID-19 is still reshaping the world economy at the moment of writing these lines. However, current effects on trade can be considered as less detrimental than in the first wave from March to May 2020. The reason is contingent on countries' capacity of adaptation to the different stages of the crisis. Countries may need to mitigate this trade shock by implementing public expenditure programs, as well as encouraging private investment. Such governmental actions may require further institutional initiatives, given the importance of the latter's sizeable effects on trade flows (e.g., [47]). Nevertheless, countrywide attention has currently shifted towards vaccines, which may determine the future formulation of policies which concentrate vaccines on a small group of producers [77]. The transition to a non-COVID-19 context is expected to depend strongly on the vaccination efforts that are being undertaken by most countries. It is fundamental for countries to remain competitive throughout the course of the COVID-19 pandemic, simultaneously rebuilding wherever possible their trade relationships.

Finally, this manuscript presents certain limitations and avenues that must be taken into consideration for future research. First, the current study only offers a preliminary impact of COVID-19 on trade, as the shock is currently ongoing. The final magnitude of the shock may be assessed once it is over. Second, we only consider aggregate trade and the impact of COVID-19 on trade may depend on the sectoral comparative advantage of each country, as shown by the previous literature [23]. Hence, we may use trade data disaggregated by sectors, although we acknowledge that sectoral trade data availability is less forthcoming than its aggregate counterpart. Third, the existence of a subset of COVID-19 stringency indicators although highly correlated with the stringency index used in this article, may be capturing measures in very specific areas.

The analysis may also be extended to services trade flows in line with [25] approach. It would also be convenient to replicate these results for the subnational level. Despite recent efforts to estimate inter and intraregional trade flows in specific areas or territories, such as the European Union [78], this data is elaborated with a considerable time lag much larger than that for country data, and such analysis is not expected to be possible in the near future. Finally, it is necessary to acknowledge the importance of firms as international trade actors since there are substantial productivity differences across exporters. For this reason, it would be convenient to extend these findings to the firm level, as studied by [79] for Colombian firms.

## Supporting information

**S1 Table. List of exporting countries.** High-income countries in bold.
(DOCX)

**S2 Table. Main descriptive statistics.**
(DOCX)

**S3 Table. Results with one lag of COVID-19 government response indicators estimated by PPML, January 2019–October 2020.** Dependent variable is trade flows. Robust standard errors in parentheses, such as *** $p<0.01$, ** $p<0.05$, * $p<0.1$. All the specifications include exporter-month, importer-month and pair fixed effects.
(DOCX)

**S4 Table. Robustness: Imports, M. Results by COVID-19 government response indicator estimated by PPML, January 2019–October 2020.** Robust standard errors in parentheses, such as *** $p<0.01$, ** $p<0.05$, * $p<0.1$. All the specifications include exporter-month, importer-month and pair fixed effects.
(DOCX)

**S5 Table. One lag of COVID-19 government response indicator.** Results by income levels. Estimated by PPML, January 2019–October 2020. Dependent variable is exports. Robust standard errors in parentheses, such as *** p<0.01, ** p<0.05, * p<0.1. Each COVID-19 indicator is estimated on a different regression but paired in the same column for the sake of brevity. All the specifications include exporter-month, importer-month and pair fixed effects. (DOCX)

**S6 Table. Robustness: Imports, M. Results by income levels and COVID-19 indicators estimated by PPML, January 2019-October 2020.** Dependent variable is imports. Robust standard errors in parentheses, such as *** p<0.01, ** p<0.05, * p<0.1. Each COVID-19 indicator is estimated on a different regression but paired in the same column for the sake of brevity. All the specifications include exporter-month, importer-month and pair fixed effects. (DOCX)

## Acknowledgments

We thank two anonymous reviewers for their useful comments, which have contributed to improving the quality of the manuscript. Comments received from attendants of XXII Conference on International Economics and XXIII Applied Economics Meetings are also gratefully acknowledged. The views expressed are purely those of the authors and cannot under any circumstances be regarded as stating an official position on the part of the European Commission.

## Author Contributions

**Conceptualization:** Javier Barbero, Juan José de Lucio, Ernesto Rodríguez-Crespo.

**Data curation:** Javier Barbero, Juan José de Lucio, Ernesto Rodríguez-Crespo.

**Formal analysis:** Javier Barbero, Juan José de Lucio, Ernesto Rodríguez-Crespo.

**Funding acquisition:** Juan José de Lucio, Ernesto Rodríguez-Crespo.

**Investigation:** Javier Barbero, Juan José de Lucio, Ernesto Rodríguez-Crespo.

**Methodology:** Javier Barbero, Juan José de Lucio, Ernesto Rodríguez-Crespo.

**Project administration:** Javier Barbero, Juan José de Lucio, Ernesto Rodríguez-Crespo.

**Resources:** Javier Barbero, Juan José de Lucio, Ernesto Rodríguez-Crespo.

**Software:** Javier Barbero, Juan José de Lucio, Ernesto Rodríguez-Crespo.

**Supervision:** Javier Barbero, Juan José de Lucio, Ernesto Rodríguez-Crespo.

**Validation:** Javier Barbero, Juan José de Lucio, Ernesto Rodríguez-Crespo.

**Visualization:** Javier Barbero, Juan José de Lucio, Ernesto Rodríguez-Crespo.

**Writing – original draft:** Javier Barbero, Juan José de Lucio, Ernesto Rodríguez-Crespo.

**Writing – review & editing:** Javier Barbero, Juan José de Lucio, Ernesto Rodríguez-Crespo.

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
