## [Decision Letter · Decision Letter 0]

11 Aug 2021

PONE-D-21-12117

Effects of COVID-19 on trade flows: measuring their impact through government policy responses

PLOS ONE

Dear Dr. de Lucio,

Thank you for submitting your manuscript to PLOS ONE. After careful consideration, we feel that it has merit but does not fully meet PLOS ONE’s publication criteria as it currently stands. Therefore, we invite you to submit a revised version of the manuscript that addresses the points raised during the review process.

The discussion should be extended, whereas the structure of the paper should be revised.

We look forward to receiving your revised manuscript.

Kind regards,

Stefan Cristian Gherghina, PhD. Habil.

Academic Editor

PLOS ONE

1. Please ensure that your manuscript meets PLOS ONE's style requirements, including those for file naming. The PLOS ONE style templates can be found at https://journals.plos.org/plosone/s/file?id=wjVg/PLOSOne_formatting_sample_main_body.pdf and https://journals.plos.org/plosone/s/file?id=ba62/PLOSOne_formatting_sample_title_authors_affiliations.pdf.

“de Lucio and Rodríguez-Crespo thank financial support from Universidad de Alcalá de Henares (UAH) and Banco Santander through research project COVID-19 UAH 2019/00003/016/001/007. De Lucio also thanks financial support from Comunidad de Madrid and UAH (ref: EPU-INV/2020/006 and H2019/HUM5761). The views expressed are purely those of the authors and may not in any circumstances be regarded as stating an official position of the European Commission.”

“de Lucio and Rodríguez-Crespo thank financial support from Universidad de Alcalá de Henares (UAH) and Banco Santander through research project COVID-19 UAH 2019/00003/016/001/007. De Lucio also thanks financial support from Comunidad de Madrid and UAH (ref: EPU-INV/2020/006 and H2019/HUM5761).”

Additional Editor Comments (if provided):

Reviewers' comments:

Reviewer's Responses to Questions

**Comments to the Author**

1. Is the manuscript technically sound, and do the data support the conclusions?

Reviewer #1: Partly

Reviewer #2: Yes

2. Has the statistical analysis been performed appropriately and rigorously? 

Reviewer #1: Yes

Reviewer #2: Yes

3. Have the authors made all data underlying the findings in their manuscript fully available?

Reviewer #1: Yes

Reviewer #2: Yes

4. Is the manuscript presented in an intelligible fashion and written in standard English?

Reviewer #1: Yes

Reviewer #2: Yes

5. Review Comments to the Author

Reviewer #1: This paper tried to figure out the impact of COVID-19 on bilateral trade flows. Authors further analyze the case by looking at the country members of regional trade agreements, governmental actions, exporter and importer country sharing identical income levels.

It would be better for the authors to give the readers more information about what policy or steps are taken during different time period. Please explain more about the different trends in Figure 1 and Figure 2.

Pay attention to the structure of your paper. Maybe the estimation strategy should be organized in the section of model description.

Although the author compared countries with different income levels, we still have no idea about which countries really suffered in the COVID-19 and what are the differences between government policy responses. More discussion or insights should be given here.

Reviewer #2: This paper examines the impact of COVID-19 on bilateral trade flows using a a gravity model of trade. The empirical analysis is designed properly. Using the monthly trade data of 68 countries, findings suggest a greater negative impact of COVID-19 on bilateral trade for countries that were members of regional trade agreements before the pandemic. The impact of COVID-19 is negative when governmental actions are considered. The article is well written.

6. PLOS authors have the option to publish the peer review history of their article (what does this mean?). If published, this will include your full peer review and any attached files.

Reviewer #1: No

Reviewer #2: No

---

## [Author Response · Author response to Decision Letter 0]

17 Sep 2021

Letter to editor

Responses to Editor and Reviewer's Reports on PONE-D-21-12117, “Effects of COVID-19 on trade flows: measuring their impact through government policy responses”

We would like to thank the Editor for offering us the opportunity to revise and resubmit our manuscript to PLOS ONE and for your interest, effort and comments on our manuscript.

All comments were very well taken, and we have tried to address them as best as

we could in preparing the revised version of the manuscript.

We organize our responses to reviewers in the same order as comments appeared in the editor letter. Further, for your convenience, we repeat comments in blue color before describing how we dealt with each of them

We have substantially revised the paper based on the comments and suggestions by the two reviewers and the Editor. In sum, in our view, the revision has led to a substantially improved paper. 

Let us summarize the main changes in the paper:

1. We have extended the discussion.

2. We have revised the structure of the paper.

3. We have updated the literature incorporating those reference that appeared since we send the paper to review, note that the topic is burgeon.

4. We fixed the format of the document (references, figures…)

These updates have substantially improved the content of the document and we thank for it. We hope you find the revision satisfactory.

Sincerely,

 

Editor Comments

Comment: “The discussion should be extended, whereas the structure of the paper should be revised.”

Authors´ response: We have done both, extend the discussions and revise the structure. 

We have widely extended the discussion mainly in section 2 “Literature review on COVID-19 and trade” and in the new section entitle “Policy responses to COVID-19: equal or unequal?” but we also have new paragraphs in sections “Results by levels of economic development” and “Conclusions” These new content focus on the policy response of government explaining better the COVID-19 related measures, in which the paper focus. We also guide the readers to additional sources of information on policy measures adopted by countries from a more general economic policy perspective (monetary, fiscal, etc.), of lower relevance in the paper.

We have a new section “Empirical analysis” At the beginning of this section we explain “This section is organized into three separate sub-sections. First, we describe the empirical model and the estimation strategy. Second, we report information on data issues. Finally, we cover country policy responses to COVID-19.” The new section merges previous sections: “Empirical model” and “Data and estimation strategy” extending content with additional information.

 

Reviewer 1 comments

Comment 1: “It would be better for the authors to give the readers more information about what policy or steps are taken during different time period. Please explain more about the different trends in Figure 1 and Figure 2.”

Comment 3: “Although the author compared countries with different income levels, we still have no idea about which countries really suffered in the COVID-19 and what are the differences between government policy responses. More discussion or insights should be given here.”

Authors´ response to both comments: Thank you for the comments, the paper is now more informative on economic policy responses. We have included additional information. First, we have redesign Figure 1 providing additional detail for countries and confidence intervals for the different indicators, we introduce explanations of the new Figure 1; for harmonization purposes we also modified figure 2. We copy here the new figure 1:

Figure 1. Evolution of the four COVID-19 policy indicators between December 2019 and October 2020 by income level for our sample of exporting countries. Source: own elaboration from [1]. Note: each point represents a country, and the concentration of countries with similar values produces darker areas. Additionally, the mean and 95% confidence bands are represented.

Second, we pay special attention to differences between high-income and low- and middle-income countries and the set of policy indicators developed by Hale et al. (2021), the most remarkable approach to measure countries´ policy responses, in line with our research objectives. Third, we suggest additional references for the interested reader in the analysis of differences between government policy responses. Apart from less noticeable changes we highly the following paragraphs:

We acknowledge that policy responses differ by country, as the impacts of COVID-19 have been strongly unequal for countries due to several reasons. First, countries have reported differences in the number of deaths, mainly attributable to the population composition. There is an increasing number of elder populations in a significant number of high-income OECD countries [2] and this group is the most vulnerable to COVID-19 (e.g., [3]). 

At the same time, countries have also implemented trade policy actions to mitigate the influence of COVID-19 on the global economy . [4] summarizes the major stylized facts during the first nine months of the pandemic . First, there was a noticeable rise in trade policy activism consisting mainly of export controls and import liberalization measures with strong cross-country variation. Second, this activism was reported to vary by country and products, where medical and food products experimented a substantial overall increase in their demand from February 2020. Third, we observe a further trade liberalization process after May 2020, where the number of liberalization measures exceeded the number of trade restrictions in medical products.

Such cross-country variation in trade policy response aligns with our expectations since, as mentioned previously, trade specialization differs by country. Accordingly, their sensitivity to the growing demand for food and medical products may vary substantially. For this reason, some countries were more resilient to COVID-19 trade shocks than other countries, as shown by the decreases observed in their trade flows. To this end, we compare trade drops for the most affected countries relative to January 2020 and their governmental response, from May 2020 to October 2020. As described by [4], countries experienced a substantial relaxation in most of their trade measures in May 2020.

For the ten countries with the largest trade drop evidence is ambivalent . On the one hand, four high-income (Macao, Mauritius, Portugal, and Slovakia) and six middle-income countries (El Salvador, Mexico, Montenegro, Guyana, Egypt, and Romania) were among the most affected countries in May 2020, with El Salvador registering the highest level of governmental response. Trade relative to January 2020 ranges from 51 to 69 percent in this period. On the other hand, we find that the number of high-income countries increased to six in October 2020 but, at the same time, differences in governmental response decreased their observed variance. Israel registered the highest level of governmental response during this month. In this case, relative trade ranges from 76 to 102, corroborating the previous finding that countries recovered rapidly from this trade shock. To sum up, despite differences in governmental response due to the impact of COVID-19 by countries, recovery can be alleged to follow similar patterns for the most affected countries.

We consider that with the modifications we provide more insights about differences between government policy responses and income levels. 

Comment 2: “Pay attention to the structure of your paper. Maybe the estimation strategy should be organized in the section of model description.”

Authors´ response: We have rearranged sections and subsections in the manuscript: we have moved data and estimation strategy to section 3, which has been renamed to “empirical analysis”. Estimation strategy has been merged with empirical model, in a new sub-section entitled “Model description and estimation strategy”. With this change, manuscript structure is more streamlined, and its soundness has also increased in comparison to the previous version.

 

Reviewer 2 

Comment 1: “This paper examines the impact of COVID-19 on bilateral trade flows using a gravity model of trade. The empirical analysis is designed properly. Using the monthly trade data of 68 countries, findings suggest a greater negative impact of COVID-19 on bilateral trade for countries that were members of regional trade agreements before the pandemic. The impact of COVID-19 is negative when governmental actions are considered. The article is well written.”

Authors´ response: We thank the reviewer for comments and effort reviewing the manuscript.

 

References

1. Hale T, Angrist N, Goldszmidt R, Kira B, Petherick A, Phillips T, et al. A global panel database of pandemic policies (Oxford COVID-19 Government Response Tracker). Nature human behaviour. 2021;5: 529-538. doi: https://doi.org/10.1038/s41562-021-01079-8.

2. OECD. Elderly population. . 2014. doi: https://doi.org/https://doi.org/10.1787/8d805ea1-en.

3. Daoust J. Elderly people and responses to COVID-19 in 27 Countries. PloS one. 2020;15. doi: https://doi.org/10.1371/journal.pone.0235590.

4. Evenett S, Fiorini M, Fritz J, Hoekman B, Lukaszuk P, Rocha N, et al. Trade policy responses to the COVID‐19 pandemic crisis: Evidence from a new data set. The World Economy. 2021. doi: https://doi.org/10.1111/twec.13119.

---

## [Decision Letter · Decision Letter 1]

27 Sep 2021

Effects of COVID-19 on trade flows: measuring their impact through government policy responses

PONE-D-21-12117R1

Dear Dr. de Lucio,

We’re pleased to inform you that your manuscript has been judged scientifically suitable for publication and will be formally accepted for publication once it meets all outstanding technical requirements.

Kind regards,

Stefan Cristian Gherghina, PhD. Habil.

Academic Editor

PLOS ONE

Additional Editor Comments (optional):

Reviewers' comments:

Reviewer's Responses to Questions

**Comments to the Author**

1. If the authors have adequately addressed your comments raised in a previous round of review and you feel that this manuscript is now acceptable for publication, you may indicate that here to bypass the “Comments to the Author” section, enter your conflict of interest statement in the “Confidential to Editor” section, and submit your "Accept" recommendation.

Reviewer #2: All comments have been addressed

2. Is the manuscript technically sound, and do the data support the conclusions?

Reviewer #2: Yes

3. Has the statistical analysis been performed appropriately and rigorously? 

Reviewer #2: Yes

4. Have the authors made all data underlying the findings in their manuscript fully available?

Reviewer #2: No

5. Is the manuscript presented in an intelligible fashion and written in standard English?

Reviewer #2: Yes

6. Review Comments to the Author

Reviewer #2: The revised version is written well. It has addressed the comments in a suitable manner. It flows well.

7. PLOS authors have the option to publish the peer review history of their article (what does this mean?). If published, this will include your full peer review and any attached files.

Reviewer #2: No

---

## [Editor Report · Acceptance letter]

5 Oct 2021

PONE-D-21-12117R1 

Effects of COVID-19 on trade flows: measuring their impact through government policy responses 

Dear Dr. de Lucio:

I'm pleased to inform you that your manuscript has been deemed suitable for publication in PLOS ONE. Congratulations! Your manuscript is now with our production department. 

Kind regards, 

on behalf of

Dr. Stefan Cristian Gherghina 

Academic Editor

PLOS ONE